



# The Common Community Physics Package (CCPP) Framework v6

Dominikus Heinzeller[1,2,3,*], Ligia Bernardet[2,3], Grant Firl[2,3,4], Man Zhang[1,2,3], Xia Sun[1,2,3], and Michael Ek[3]

[1]University of Colorado Boulder, Cooperative Institute for Research in Environmental Sciences (CIRES), Boulder, CO, 80309, USA
[2]National Oceanic and Atmospheric Administration (NOAA), Earth System Research Laboratories, Global Systems Laboratory (GSL), Boulder, CO, 80305, USA
[3]Developmental Testbed Center (DTC), Boulder, CO, 80301, USA
[4]Colorado State University, Cooperative Institute for Research in the Atmosphere (CIRA), Fort Collins, CO, 80521, USA
[*]Now at: University Corporation for Atmospheric Research (UCAR), Joint Center for Satellite Data Assimilation (JCSDA), Boulder, CO, 80301, USA

**Correspondence:** Ligia Bernardet (ligia.bernardet@noaa.gov)

**Abstract.** The Common Community Physics Package (CCPP) is a collection of atmospheric physical parameterizations for use in Earth system models and a framework that couples the physics to a host model's dynamical core. A primary goal for this effort is to facilitate research and development of physical parameterizations and experimentation with physics-dynamics coupling methods, while simultaneously offering capabilities for use in numerical weather prediction (NWP) operations. The

CCPP Framework supports configurations ranging from process studies to operational NWP as it enables host models to assemble the parameterizations in flexible suites. Framework capabilities include variability of scheme call order, ability to group parameterizations for calls in different parts of the host model allowing intervening computation or coupling to additional components, options to call some parameterizations more often than others, and automatic variable transformations.

The CCPP Framework was developed by the Developmental Testbed Center and is distributed with a single-column model
that can be used to test innovations and to conduct hierarchical studies in which physics and dynamics are decoupled. It is also an integral part of the Unified Forecast System, a community-based, coupled, comprehensive Earth modeling system designed to support research and be the source system for NOAA's operational NWP applications. Finally, the CCPP Framework is under various stages of adoption by a number of other models in the wider community.

## 1 Introduction

The existence of shortcomings in the representation of physical processes has been identified as one of the primary sources of errors in numerical weather prediction (NWP) models, with other contributing factors being the imperfect specifications of initial conditions and inaccuracies introduced by the dynamical core, for example (Bauer et al., 2015; Du et al., 2018). In a NWP model, many physical processes are accounted for by often oversimplified parameterizations rather than as a result of strictly self-consistent consequences of the (nominally nonlinear and reactive) fluid dynamics both mediating and underly-

ing weather systems (adapted from the glossary by American Meteorological Society, 2022). Therefore, the improvement in forecasts from NWP models hinges on the continuous advancement of physical parameterizations in concert with increases in





numerical accuracy required to accommodate realistic transport and scale interactions as well in spatio-temporal resolutions of observations and their correct assimilation.

Examples of processes represented by the physical parameterizations in a NWP model are radiation, moist physics, vertical mixing, and interactions between the atmosphere and the underlying surface (Stensrud, 2007). State-of-the-art NWP models employ parameterizations of diverse complexity and may include representation of chemical processes that impact air composition and feed back on meteorological processes (Ahmadov et al., 2017). Parameterizations are typically used in sets called suites, which are constructed using compatible tasks that interact, ideally, consistently, often in addition to or with the aid of intermediate calculations. An important example of inter-parameterization communication is cloud-radiation feedback, which demands that the condensed water produced by the microphysics, macrophysics, convection, and planetary boundary layer parameterizations be communicated to the radiation parameterization to modify the radiative fluxes due to the presence of clouds (Črnivec and Mayer, 2019; Han et al., 2017; Bourgeois et al., 2016).

Parameterizations in modern NWP models are sophisticated and are typically the result of many years of work by a sizable number of subject matter experts. A suite may evolve through the incremental improvement in a parameterization, with a single code base being augmented to include the representation of additional processes (Han and Bretherton, 2019) or discretely, through the substitution of a parameterization with a distinct, more advanced, code base (Ukkonen et al., 2020). Therefore, it is crucial to have a software framework that can support collaborative and flexible development.

To meet this demand, the Developmental Testbed Center (DTC) has spearheaded the development of the Common Community Physics Package (CCPP), a model-agnostic collection of codes containing atmospheric physical parameterizations (CCPP Physics) along with a framework that connects the physics to host models (CCPP Framework). It is distributed with a Single-Column Model (CCPP SCM), a simple host model that can be used with CCPP-compliant physics. This initiative is part of a broader effort to develop and improve the Unified Forecast System (UFS; Jacobs, 2021), a "community-based, coupled, comprehensive Earth modeling system [that spans] local to global domains and predictive time scales from sub-hourly analyses to seasonal predictions [and] is designed to support the weather enterprise and to be the source system for NOAA's operational NWP applications" (Unified Forecast System - Steering Committee (UFS-SC) and Writing Team, 2021). The CCPP Physics contains the parameterizations that are used operationally in the atmospheric component of the UFS Weather Model (UFS Atmosphere), as well as parameterizations that are under development for possible transition to operations in the future (Zhang et al., 2020). The CCPP aims to support the broad community while simultaneously benefiting from the community. Parameterizations have been contributed by a number of scientists from various organizations, creating an ecosystem in which the CCPP can be used not only by operational centers to produce operational forecasts, but also by the research community to conduct investigation and development (see, for example, He et al., 2021). The CCPP provides the means for the necessary partnership and collaboration required to ensure that innovations created and effectively tested by the research community can be funneled back to the operational centers for further improvement of the operational forecasts.

There have been six major public releases of the CCPP thus far, starting with the v1 release in March of 2018. The latest version is CCPP v6, released in June 2022 (Firl et al., 2022). The CCPP is a component in the public releases of the UFS Medium-Range Weather Application v1.1 (UFS Community, 2021) and the UFS Short-Range Weather Application (SRW



App) v2.0 (UFS Community, 2022), and has been adopted by NOAA's National Weather Service as the physics infrastructure for all upcoming operational implementations of the UFS (Unified Forecast System Steering Committee and Writing Team 2021). It is targeted for transition to operations (Tallapragada et al., 2022) in 2023 as a component of the Hurricane Analysis
and Prediction System (HAFS) and in 2024 as part of the Global Forecast System (GFS), Global Ensemble Forecast System (GEFS), and Rapid Refresh Forecast System (RRFS). In addition to the CCPP SCM and the UFS Atmosphere, the CCPP is being used in an experimental version of the Navy Environmental Prediction System Using the NUMA Core (NEPTUNE; Doyle et al., 2022), where NUMA stands for the Non-hydrostatic Unified Model of the Atmosphere. Furthermore, NCAR is investing in advancements for the CCPP Framework with the intention of adopting it as the physics interface for future
versions of their System for Integrated Modeling of the Atmosphere (SIMA; Davis et al., 2019; Gill et al., 2020). The CCPP Framework is also part of the Multi-Scale Infrastructure for Chemistry and Aerosols (MUSICA), the NCAR-driven next-generation community infrastructure for research involving atmospheric chemistry and aerosols (Pfister et al., 2020).

To enable the community involvement demanded by the aforementioned collaborations, the CCPP was established with several considerations: interoperability (the ability to run a given physics suite in various host models), portability, extensibility,
computational efficiency, and ease-of-use. This effort builds on previous quests for physics interoperability, which started as early as the 1980s with the establishment of the so-called Kalnay rules (Kalnay et al., 1989) that outline coding standards to facilitate the exchange of parameterizations among models. These rules were later revised by members of the multi-agency National Earth System Prediction Capability (ESPC; now ICAMS) Physics Interoperability committee to reflect advances in computational hardware and software used for operational NWP (Doyle et al., 2015). The CCPP further evolves earlier efforts
in interoperability for physics, such as the noteworthy implementation of an infrastructure for interoperability is the Interoperable Physics Driver (IPD) devised by the NOAA Environmental Modeling Center (EMC), and later augmented by the NOAA Geophysical Fluid Dynamics Laboratory (GFDL), to run the physics used in the GFS in other host models (Whitaker et al., 2017). It complements similar efforts in other aspects of Earth system modeling, such as those aimed at enabling interoperability of component models (i. e., the atmospheric model, the ocean model) among various hosts via the standardization of
interfaces (Theurich et al., 2016; Wieters and Barbi, 2019) and of physical constants (Chen et al., 2020).

This paper focuses on the CCPP Framework and provides an overview of the technical approach (Section 2), a description of the implementation (Section 3), an example of CCPP use in various host models (Section 4), and an outlook for the package (Section 5).

## 2 Design

The design of the CCPP relies on standardization and automation, the latter requiring the former for an efficient implementation. All CCPP-compliant parameterizations have up to five entry point subroutines, corresponding to the model initialization, timestep initialization, timestep execution (time integration), timestep finalization, and model finalization phases of the parameterization. Each entry point subroutine is accompanied by metadata describing the variables that are passed to and from that subroutine, including attributes such as standard name, long name, intent, units, type, rank, and activity status— circum-





stances under which variables are used if they are optional. Similarly, the host model contains metadata about variables that are available to be used by the parameterizations. The CCPP Framework compares and matches the standard names, much like a database key, of variables requested by the parameterizations against those supplied by the host model, making it possible for a given parameterization to be usable by various hosts. The CCPP Framework can convert units between requested and supplied variables, thus reducing the need for developers to build converters into their parameterizations. The Framework is further capable of handling blocked host model data that is used for improving the run time performance of the physics calls by processing them in parallel with multiple threads. It also provides debugging features that assist developers in their efforts by automatically checking variable allocations as well as the size and dimensions of variables.

The variable metadata that accompany the parameterizations are also used to generate aspects of the Scientific Documentation (Zhang et al., 2022). The CCPP employs the Doxygen software (van Heesch, 2022) to parse inline comments and additional information, such as parameterization descriptions, figures, and bibliography. This data is combined with the information from the metadata tables to generate documentation in HyperText Markup Language (HTML) format.

The CCPP Framework has access to a pool of compliant parameterizations contained in the CCPP Physics. The library can contain multiple parameterizations for different physical processes that a host model may execute during the physics calls. The choice of parameterizations (referred to as a suite) that will be invoked at runtime, and the order in which they will be executed, are defined through a Suite Definition File (SDF). The SDF is organized into groups of parameterizations, which are subsets of a suite that can be called from different locations in the host model. In other words, the physics suite does not have to be executed in a single step, but can be interspersed with calls to other parts of the host model, such as dynamics and coupling with external components. In models that have shorter timesteps for calls to dynamics than to physics, this CCPP capability enables the use of *fast* and *slow* physics, that is, enables calling selected parameterizations from the dynamical core and executing them more often than the rest of the physics suite.

Through specifications in the SDF, the CCPP Framework allows the subcycling of parameterizations, that is, executing one or more parameterizations more than once in a physics timestep. Subcycling can be used to increase the stability of the model and enable larger timesteps by reducing the magnitude of the tendencies that need to be applied during each update of the model state variables. Subcycles can also be used to emulate an iteration, and a set of two subcycles can be used to set up a first guess and correction step for a parameterization. It is important to note that subcycling of schemes is only available in the run phase (see Section 3.2 for more information on the CCPP phases).

The CCPP Framework can be considered a data broker that only plays a role at model build time, but is not part of the actual model executable. At build time, one or more SDFs are provided and the Framework is invoked to check consistency between variables requested by and supplied to the physics and to auto-generate calling interfaces for each SDF (the physics and suite interface code, also referred to as *caps*). The framework further auto-generates the application programming interface (API) that the host model calls to connect to the physics through a host model cap. At runtime, a SDF must be chosen to determine the suite that will be actually used in the run, which typically happens during the model configuration and initialization. Figure 1 shows the general architecture of a modeling system that employs the CCPP, and Section 3 provides details on the CCPP technical implementation.



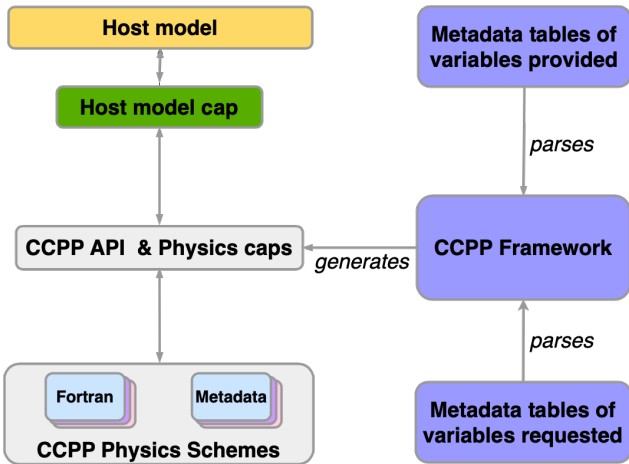

**Figure 1.** Architecture of the CCPP and its connection to a host model. The CCPP Framework auto-generates the CCPP API and physics caps that connect the schemes in CCPP Physics with the host model through a host model cap/CCPP driver.

While the original design and current use of the CCPP are centered on physics in atmospheric models, the system is general enough to be used in various host modeling systems, including complex coupled Earth system models, and for different types of processes. For instance, a coupled ocean-atmosphere-chemistry model could call CCPP from the atmospheric component, the ocean component, and the chemistry component to execute atmospheric physics parameterizations, ocean physics parameterizations, and chemical processes, respectively. It should be noted that currently all CCPP-compliant physical parameterizations and all host models using CCPP are written in Fortran, which is traditionally the dominant programming language in NWP. The concept and design of the CCPP Framework do not preclude host models or schemes using other programming languages (e. g., C/C++), but currently these are not supported by the CCPP Framework.

## 3 Implementation

Designed as an infrastructure for connecting physical parameterizations to atmospheric models, the CCPP Framework requires a host model for both development and testing. The supported host models range from basic stubs that can be used to test the framework itself to fully coupled three-dimensional Earth system models that contain multiple connections to the CCPP Physics through the auto-generated code from the CCPP Framework. This section describes in detail the implementation of the CCPP in two atmospheric host models, the CCPP SCM and the UFS, based on the design requirements outlined in Section 2. The two models will be discussed in detail in Section 4.

### 3.1 Supported parallelism

Computational efficiency is an important aspect of the design of the CCPP. The system must deliver the necessary performance for operational applications while at the same time providing options for flexibility. CCPP supports Message Passing Interface



**Table 1.** Supported phases during a model run. The third column contains information on how blocked data structures are handled in the different phases.

| Phase | Purpose | Blocked/chunked data supported? |
|---|---|---|
| `init` | Initialize physics: read/compute lookup tables, set runtime options | No, requires access to all data |
| `timestep_init` | Initialize timestep: update time, solar insolation, lookup table data | No, requires access to all data |
| `run` | Integrate physics forward | Yes |
| `timestep_finalize` | Finalize timestep: compute statistics and diagnostic tendencies | No, requires access to all data |
| `finalize` | Finalize physics: deallocate variables, close files | No, requires access to all data |

(MPI) task parallelism, OpenMP threading, and hybrid MPI+OpenMP parallelism. To accommodate the different requirements of physical parameterizations and host modeling systems, the implementation of the CCPP Framework and Physics is based
on the two following paradigms:

1. Physics are column-based and there can be no communication between neighboring columns in the physical parameterizations during the time integration phase (also referred to as timestep execution or run phase).

2. The physics and timestep initialization and finalization phases cannot be called by multiple threads in parallel.

With the above requirements in mind, the following limitations apply:

1. MPI communication is only allowed in the physics during the physics initialization, timestep initialization, timestep finalization, and physics finalization phases. The parameterizations must use the MPI communicator provided by the host model as an argument to the physics schemes.

2. The time integration (run) phase can be called by multiple threads in parallel. Threading inside the physics is allowed in every phase, but the parameterizations must use the number of available OpenMP threads provided by the host model as
an argument. It is the responsibility of the host model to handle any synchronization of MPI tasks or threads.

### 3.2 CCPP phases

The CCPP Framework supports five phases in a model run, which are summarized in Table 1. The table further contains information on how blocked data structures are handled in the individual phases. With exception of the time integration (run) phase, blocked data structures must be combined into contiguous data such that a physics scheme has access to all data that
an MPI task owns. The need to have access to all data and the limitations on OpenMP threading described in the previous section are a result of potential file Input/Output (I/O) operations, computations of statistics such as minimum values of a variable, interpolation of lookup table data (e. g., ozone concentration from climatology), and global communication during those phases.





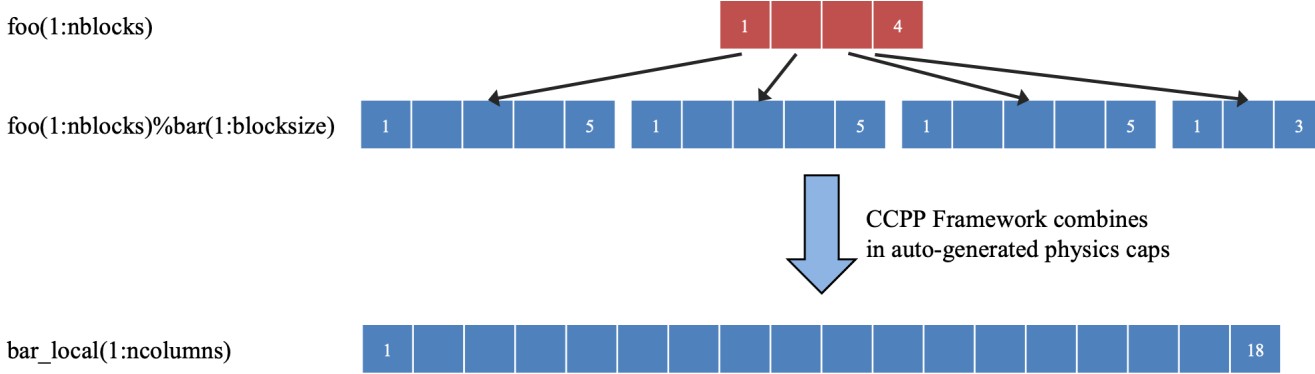

**Figure 2.** The CCPP Framework automatically combines blocked data structures into contiguous arrays during certain phases of the physics calls, as shown here for the variable `bar` that is part of derived data type `foo`, which is split up into four blocks.

**Listing 1:** Auto-generated code that combines the blocked data structure `foo(:)%bar(...)` from Figure 2 into a contiguous array. The Fortran syntax highlighting represents keywords in red, user expressions in blue, and comments in gray.

```fortran
allocate(bar_local(1:ncolumns))
ib = 1
do nb=1,nblocks
  bar_local(ib:ib+blocksize(nb)-1) = foo(nb)%bar
  ib = ib+blocksize(nb)
end do
call myscheme_init(bar=bar_local)
ib = 1
do nb=1,nblocks
  foo(nb)%bar = bar_local(ib:ib+blocksize(nb)-1)
  ib = ib+blocksize(nb)
end do
deallocate(bar_local)
```

The CCPP Framework is capable of combining blocked data structures into contiguous arrays in the auto-generated physics caps. The CCPP metadata plays an important role in determining whether such operations are required, how the data needs to be combined, and whether a variable needs to be skipped because it is inactive and may not be allocated or initialized properly (see Section 3.5 for details). Figure 2 and Listing 1 illustrate this capability.





### 3.3 CCPP-compliant schemes

CCPP-compliant parameterizations are Fortran modules with a number of requirements on formatting, naming and coding
standards, and that are accompanied by a metadata file. The name of the Fortran source file should, but does not have to,
match the name of the Fortran module, and the metadata file must match the name of the Fortran source file and carry the
extension `.meta`. The Fortran module name defines the name of the CCPP scheme and of the entry point subroutines. For
example, a Fortran module/scheme with name `myscheme` can have up to five entry points, one for each of the CCPP phases,
with names `myscheme_init`, `myscheme_timestep_init`, etc. Each CCPP entry point must be preceded by CCPP
metadata hooks that instruct the code generator to parse the associated metadata file for the metadata for the variables used in
the argument list to the particular subroutine/entry point call. As mentioned earlier in Section 2, the concept and design of the
CCPP Framework do not preclude host models or schemes using other programming languages, but currently all host models
and all CCPP-compliant parameterizations are written in Fortran.

The full set of requirements for CCPP-compliant parameterizations is described in the CCPP technical documentation
(Bernardet et al., 2022, Chapter 2). In brief, CCPP-compliant parameterizations should follow modern Fortran programming
standards, be written in Fortran 90 free-form, use `implicit none`, avoid the use of `goto` statements, and use named end
constructs for modules and subroutines to support the CCPP Framework parser. Neither Fortran `common` blocks nor the local
definition or importing of physical constants or functions from other physical parameterizations are allowed. Physical con-
stants must be passed to the physics schemes via the argument list. Each CCPP scheme may define its own dependencies (i. e.,
Fortran, C, C++, . . . modules) that are required for the scheme, as long as these do not depend on the presence of other schemes
and vice versa. One exception to this rule is the definition of floating point kinds, which are currently provided in the form of
a Fortran module that resides in CCPP Physics and that can be imported by the physics schemes and the host model.

Each entry point that is in use (i. e., that is not empty) and that is preceded by CCPP metadata hooks must accept two
mandatory variables that are used by the CCPP error handler: an error code and an error message. CCPP schemes are not
allowed to stop the model execution, and writing to `stdout` and `stderr` is discouraged. In the event of an error, a scheme
must set the error code to a non-zero value, assign a meaningful error message and return control to the caller. Listing 2 presents
a complete CCPP-compliant parameterization side-by-side with its metadata, which will be discussed in detail in Section 3.5.1.

### 3.4 Host model variable definitions

The host model is responsible for declaring, allocating, initializing and, if applicable, resetting all variables that are required to
195 execute the physical parameterizations. Each variable passed to the physics must be a Fortran standard variable (real, integer,
. . . ) of rank zero or greater, or a derived data type (DDT) that is defined by the receiving physical parameterization itself. DDTs
defined by the host model cannot be passed to the physical parameterizations, as doing so would create a dependency of the
CCPP Physics on a particular host model. However, the Fortran standard variables or physics DDTs can be constituents of
host model DDTs, and they can be statically or dynamically allocated, or pointers. Host model variables must be accessible
from a module, because the auto-generated CCPP API (see Section 3.7) imports the variables (in case of Fortran standard





**Listing 2:** A complete CCPP-compliant parameterization with Fortran source code (left) and corresponding metadata (right). Details for allowed metadata descriptions can be found in Bernardet et al. (2022, Chapter 2).

| `myscheme.F90` | `myscheme.meta` |
|---|---|

```fortran
module myscheme

  use kind_defs, only: kind_phys
  implicit none

  contains

    ! Inactive entry points can be omitted

!> \section arg_table_myscheme_run Argument Table
!! \htmlinclude myscheme_run.html
!!
  subroutine myscheme_run (bar, errmsg, errflg)

    ! arguments
    real(kind_phys),  intent(inout) :: bar(:,:)
    character(len=*), intent(out)   :: errmsg
    integer,          intent(out)   :: errflg

    ! local variables
    ! add your local variables here

    ! initialize CCPP error handling variables
    errmsg = ''
    errflg = 0

    ! initialize intent(out) variables

    ! add your code here

    ! in case of errors, set errflg != 0, assign
    ! a meaningful message to errmsg and return

  end subroutine myscheme_run

end module myscheme
```

```
[ccpp-table-properties]
  name = myscheme
  type = scheme
  dependencies = kind_defs.F90

[ccpp-arg-table]
  name = myscheme_run
  type = scheme
[bar]
  standard_name = std_name_for_bar
  long_name = description of bar
  units = valid unit for bar
  dimensions = \
       (horizontal_loop_extent, \
        vertical_layer_dimension)
  type = real
  kind = kind_phys
  intent = inout
[errmsg]
  standard_name = ccpp_error_message
  long_name = error handling message
  units = none
  dimensions = ()
  type = character
  kind = len=*
  intent = out
[errflg]
  standard_name = ccpp_error_code
  long_name = error handling code
  units = 1
  dimensions = ()
  type = integer
  intent = out
```



variables) or the parent DDTs and passes them to the auto-generated physics caps. Listing 3 provides an example of a host model Fortran module and metadata that define variables for use by the physics. The host model metadata will be discussed further in Section 3.5.2.

Variables that are in use by the physical parameterizations selected for use at runtime (see Section 3.6) must be allocated and initialized by the host model. Depending on the choice of physics or their runtime configuration, variables may be left unallocated when being passed to the physical parameterizations. These variables are considered to be inactive and must have the corresponding metadata attribute `active` (see Section 3.5.3) set accordingly in the host model metadata so that the CCPP Framework skips any variable transformations that would lead to an invalid memory access. This mechanism was created to reduce the memory footprint of the application. A consequence of potentially unallocated host variables is that dummy argument variable arrays within schemes should be declared as assumed-shape to avoid compilation errors. Some variables need to be reset at certain times, such as accumulation "buckets" for diagnostics, and the host model is required to perform these actions. Likewise, the CCPP Framework does not provide a mechanism for writing diagnostics or restart data to disk – it is the responsibility of the host model to know which data to write to disk and when.

To ensure a consistent set of physical constants for use by all physical parameterizations, these constants must be defined by the host model and passed to the physics via the argument list, in other words, they are treated like normal variables. The host model in this case imports these kind/DDT definitions from the physics to allocate the necessary variables. In other words, while interoperability considerations preclude a dependence of the CCPP Physics on the host model, the need to manage the memory and initialization of variables in use by the physics creates a dependency of the host model on the CCPP Physics.

## 3.5 Metadata

The metadata are the essential pieces of information that allow the code generator (presented in Section 3.7) to connect the individual physics parameterizations to the host model. The implementation of the metadata follows a relaxed Python config-style format, except that a CCPP metadata file can contain the same keyword more than once. As mentioned before, the CCPP requires an associated metadata file for each scheme's Fortran source file containing the scheme entry points to be located in the same directory, with the same filename and extension `.meta`. Similarly, for each Fortran source file on the physics and host model side that contains the kind, type or variable definitions needed to run the CCPP Physics, a metadata file is required in the same directory, with the same filename and extension `.meta`. The two different types of metadata are referred to as *scheme metadata* and *host metadata* and will be described in detail in the following sections.

### 3.5.1 Scheme metadata

The role of metadata for CCPP-compliant schemes is to describe in detail the variables that are required to call a scheme for each of the phases described in Section 3.3. Each CCPP scheme has a set of metadata that consists of a header section `[ccpp-table-properties]` and individual sections labeled with `[ccpp-arg-table]`, one for each phase in use, followed by a list of all variables and their attributes required to call the scheme phase. The header section must contain the name of the scheme and a `type = scheme` option that tells CCPP that this is scheme metadata. It can also contain one or





more `dependencies = ...` options, and each of them can be a comma-separated list of dependencies for this scheme. A
235 further `relative_path` option can be used to append a relative path to all dependencies listed in the header section.

Each `[ccpp-arg-table]` section repeats the `type = scheme` and lists the name of the CCPP entry point, which
consists of the name of the scheme and the CCPP phase, connected by an underscore. Each variable is then described in a
separate section with the local name of the variable as the section identifier, see Listing 2, right column, for an example. With
the exception of the `kind` option, all attributes shown in Listing 2 for variable `bar` are mandatory. It is recommended, although
not yet enforced, that the order of variables in the metadata file matches the argument list of the Fortran subroutine for ease of
debugging.

### 3.5.2  Host model metadata

The host model metadata, as well as kind and type definitions in CCPP Physics, are used to define kinds, DDTs and variables
that are used by the host model to execute the chosen physics schemes. The host model variables and the scheme variables
are paired by their standard names. The CCPP Framework code generator compares the variables provided by the host model
against the variables requested by the physics schemes in the list of suites provided at compile time (see Section 3.7 for details).
In case of a mismatch, the Framework throws an error. Likewise, variables have to match in all their attributes except their local
names (due to differing variable scopes), long names and units (see Section 3.5.3).

The construction of metadata information in the host model is complicated by the fact that the metadata alone must provide
the CCPP Framework code generator with enough information on where to find the variables required by the physical parame-
terizations. The metadata table in Listing 3 illustrates this. While it is straightforward to declare Fortran standard variables on
a module level (example `baz` in Listing 3), a variable that is a constituent of a DDT requires a metadata table for the DDT,
and a definition and an instance of the DDT in the correct module metadata following the exact syntax in Listing 3. The DDT
type `foo_type` in this example is used to store blocked data of a two-dimensional array `bar` and is defined and allocated in
module `physics_var_defs`. Array `bar` is of kind `kind_phys`, which is defined in and imported from CCPP Physics.




**Listing 3:** Host model variable declaration and associated metadata. The definition of `kind_phys` is imported from the CCPP Physics, which contains `kind_defs.F90` and `kind_defs.meta`. Details for allowed metadata descriptions can be found in Bernardet et al. (2022, Chapter 6).

---

**physics_var_defs.F90**

```fortran
  module physics_var_defs

    use kind_defs, only: kp => kind_phys
    implicit none

!> \section arg_table_physics_var_defs
Argument Table
!! \htmlinclude physics_var_defs.html
!!
    integer, parameter :: baz = 1
!> \section arg_table_foo_type Argument Table
!! \htmlinclude foo_type.html
!!
    type foo_type
       real(kp), pointer :: bar(:,:)
    contains
       procedure :: create => foo_create
       procedure :: reset => foo_reset
    end type foo_type
    type(foo_type), allocatable :: foo(:)
  contains

  subroutine foo_create (foo,ncol,nlev)
     class(foo_type) :: foo
     integer, intent(in) :: ncol, nlev
     allocate(foo%bar(ncol,nlev))
     call foo%reset()
  end subroutine foo_create

  subroutine foo_reset (foo)
     class(foo_type) :: foo
     foo%bar = 0.0_kp
  end subroutine foo_reset

  end module physics_var_defs
```

**physics_var_defs.meta**

```
[ccpp-table-properties]
  name = physics_var_defs
  type = module
  relative_path = /path/to/physics
  dependencies = kind_defs.F90

[ccpp-arg-table]
  name = physics_var_defs
  type = module
[baz]
  standard_name = std_name_for_baz
  long_name = description for baz
  units = 1
  dimensions = ()
  type = integer
[foo_type]
  standard_name = foo_type
  long_name = definition of type foo_type
  units = DDT
  dimensions = ()
  type = foo_type
[foo(ccpp_block_number)]
  standard_name = foo_type_instance
  long_name = instance of derived type foo_type
  units = DDT
  dimensions = ()
  type = foo_type

[ccpp-table-properties]
  name = foo_type
  type = ddt
  relative_path = /path/to/physics
  dependencies = kind_defs.F90
...
```





**Listing 3 (continued).**

```
...
[ccpp-arg-table]
  name = foo_type
  type = ddt
[bar]
  standard_name = std_name_for_bar
  long_name = description of bar
  units = valid unit for bar
  dimensions = \
        (horizontal_loop_extent, \
         vertical_layer_dimension)
  type = real
  kind = kind_phys
```

---

**kind_defs.F90**                                    **kind_defs.meta**

```
  module kind_defs                          [ccpp-table-properties]
                                              name = kind_defs
    implicit none                            type = module
    private                                  dependencies =
    public :: kind_phys
                                            [ccpp-arg-table]
!> \section arg_table_kind_defs Argument Table    name = kind_defs
!! \htmlinclude kind_defs.html                     type = module
!!                                          [kind_phys]
    integer, parameter :: kind_phys = 8      standard_name = kind_phys
                                              long_name = definition of kind_phys
                                              units = none
  end module kind_defs                       dimensions = ()
                                              type = integer
```

---





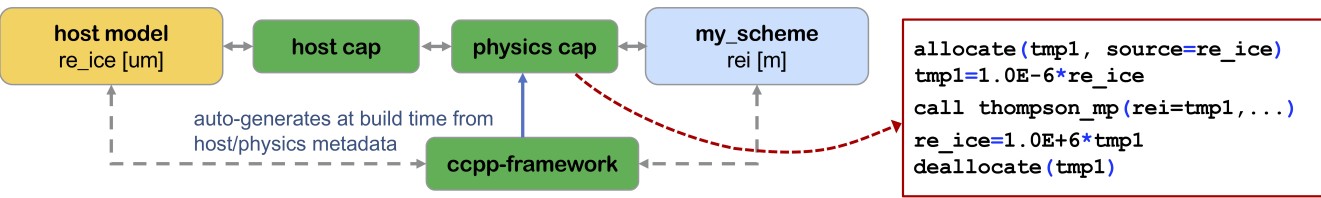

**Figure 3.** Example for automatic unit conversions in the auto-generated physics cap.

### 3.5.3 Variable attributes

Each variable is defined by a local name and a set of variable attributes. The local name can be different in each of the physics schemes and in the host model. The physics schemes and the host model share the following variable attributes:

**standard_name** The standard name is key for matching variables between the host model and a physics scheme. It must therefore be unique in the host model and within each scheme. To secure interoperability, the CCPP standard names are derived from a clear set of rules. These rules follow and extend the NetCDF Climate and Forecast (CF) metadata conventions (Hassell et al., 2017) and are defined in the CCPPStandardNames GitHub repository (Earth System Community Modeling Portal, 2022a). The repository also contains information about acronyms and units of variables, as well as a table with all standard names that are currently in use in any of the CCPP-compliant physics. Users developing new schemes or importing schemes into CCPP must consult the existing standard names before creating new standard names.

**long_name** The long name contains a brief description of the variable. This can be used to further clarify the purpose of this variable. The long name can be different in each of the physics schemes and the host model to provide the necessary clarification in the given context.

**units** The variable units must follow the definitions in the UDUNITS package (Unidata, 2022). If a host model variable and a physics variable matched by their standard name differ in their units, the CCPP Framework is capable of performing automatic unit conversions in the auto-generated physics caps. Unit conversions are implemented in the CCPP Framework code generation scripts. An error will be thrown when running the CCPP Framework in the code generation step if a particular unit conversion is requested but not implemented. In the event of such an error, one may define the appropriate unit conversion in the CCPP Framework or otherwise manually write conversions in the scheme's code to ensure unit consistency. Figure 3 provides an example of an automatic unit conversion for cloud effective radii between a host model (using micrometer [um]) and a physics scheme (using meter [m]).

**dimensions** The dimensions of a variable are listed in the form of CCPP standard names and are explained further in Section 3.5.4. The variable's dimensions must be the same for the host model and the physics schemes. The only exception are `horizontal_dimension` or `horizontal_loop_extent`, depending on the CCPP phase and whether blocked data structures are used (see Section 3.5.4 for details).





**Listing 4:** Metadata for variable `surface_snow_area_fraction_over_ice` in the host model and a physics scheme.

| `physics_var_defs.meta` (host model metadata) | `my_scheme.meta` (physics scheme metadata |
|---|---|

```
[sncovr_frac_ice]
  standard_name = \
        surface_snow_area_fraction_over_ice
  long_name = surface snow area fraction ice
  units = frac
  dimensions = (horizontal_loop_extent)
  type = real
  kind = kind_phys
  active = (control_for_land_surface_scheme \
   == identifier_for_ruc_land_surface_scheme)
```

```
[sncovr_ice]
  standard_name = \
        surface_snow_area_fraction_over_ice
  long_name = ice surface snow area fraction
  units = frac
  dimensions = (horizontal_loop_extent)
  type = real
  kind = kind_phys
  intent = in
```

**type** The type of a variable can either be a Fortran standard type (`integer`, `real`, `logical`, `complex`) or a DDT (`ddt`). The use of DDTs in a physics scheme is possible as long as the DDT itself is defined in the physics scheme, not in the host model. The type of a variable must be the same for the host models and the physics scheme.

**kind** This is an optional attribute that is only required for Fortran variables that use a kind specifier. The kind of a variable must match between the host model and the schemes.

Variable metadata for physics schemes in addition contain an **intent** attribute, which is identical to the definition in Fortran (`in`, `inout`, `out`). It is important to use the correct value for this attribute, because the CCPP Framework may omit certain variable transformations in the auto-generated caps based on its value. For example, if a variable is declared as `intent = in`, automatic unit conversions will only be performed on the temporary variable before entering the scheme, and the reverse operation after returning from the scheme will be omitted. Likewise, for a variable declared as `intent = out`, the temporary variable for combining blocked data structures will be allocated but left uninitialized before entering the physics scheme, since the scheme is expected to overwrite the contents of this variable completely. The `intent` information is also used by the variable tracker described in Section 3.7.

The host model metadata has the additional, optional attribute **active**, which contains a Fortran logical condition expressed in CCPP standard names that describes whether a variable is allocated or not. The default value is `.true.`, which means that the variable is always allocated. If a variable is inactive, the auto-generated code will skip any operation on the variable in the auto-generated caps (e. g., unit conversions). Listing 4 contains examples for a variable defined in a host model and in a physics scheme.





### 3.5.4 Variable dimensions

Variable dimensions can be coordinate dimensions or indices. An example for an index dimension is a particular tracer in an array of tracers. For performance reasons, the index dimensions for tracer arrays are typically the outermost (i. e., rightmost) dimension in Fortran. Individual tracers can then be passed efficiently to physics schemes as a contiguous slice of the tracer array in memory. Coordinate dimensions for CCPP Physics schemes consist of one horizontal dimension and one vertical dimension, with the convention that the horizontal dimension is the innermost (i. e., leftmost) dimension in Fortran.

The choice of only one horizontal dimension was made because many modern host models use unstructured or irregular meshes, and because any two horizontal dimensions can be passed to physics schemes using one horizontal dimension only, as long as the horizontal dimensions are the innermost (i. e., leftmost) dimensions for both the host model and the physics schemes. It should be noted that the CCPP Framework currently does not support array transformations to accommodate host models that use the vertical dimension as the innermost dimension. For such a host model, a manually written host model cap

must transform all arrays before entering and after returning from the physical parameterization. The CCPP standard names for vertical dimensions distinguish between layers (full levels), where `vertical_layer_dimension` is used, and interfaces (half levels), where `vertical_interface_dimension` is used. Additional qualifiers can be appended, for example as in `vertical_interface_dimension_for_radiation`.

The host model must define two variables to represent the horizontal dimension in the metadata. The variable with standard

name `horizontal_dimension` corresponds to all columns that an MPI task owns (or simply all horizontal columns when no MPI parallelism is used). The variable with standard name `horizontal_loop_extent` corresponds to the size of the chunk of data that is sent to the physics in the run phase (see Section 3.2). In the simplest example, the host model passes all horizontal columns of an MPI task at once, and the variables `horizontal_dimension` and `horizontal_loop_extent` are identical. The more complicated scenario is illustrated in Listing 3, where the host model defines a vector `foo` of type

`foo_type` that contains blocks of a two-dimensional array `bar`. In the run phase, the physical parameterizations are called for one block at a time (although possibly in parallel using OpenMP threading). Here, `horizontal_loop_extent` corresponds to the block size, and the sum of all block sizes equals `horizontal_dimension`. In either of these cases, the convention is to use `horizontal_loop_extent` as the correct horizontal dimension for host model variables.

The CCPP physics schemes have no knowledge about the storage scheme of the host model or about how many hori-

325 zontal columns are passed. The correct standard name in the scheme metadata depends on the CCPP phase. As discussed in Section 3.2, all phases except the run phase expect the entire data that an MPI task owns. The correct standard name for the horizontal dimension in the `init`, `timestep_init`, `timestep_finalize` and `finalize` phases is thus `horizontal_dimension`. For the `run` phase, the correct standard name is `horizontal_loop_extent`. Thus, the same variable must use a different standard name in its horizontal dimension attribute depending on the CCPP phase.

These rules may sound confusing at first, but they allow the CCPP Framework to handle different strategies for memory allocation and thread parallelism while securing interoperability and being able to reliably catch errors in the metadata.





**Listing 5:** A simple Suite Definition File `suite_ModelX_v1.xml` containing two groups.

```xml
<?xml version="1.0" encoding="UTF-8"?>

<suite name="ModelX_v1" version="1">
  <group name="groupA">
    <subcycle loop="1">
      <scheme>interstitial_scheme_1</scheme>
      <scheme>primary_scheme_1</scheme>
      <scheme>interstitial_scheme_2</scheme>
    </subcycle>
  </group>
  <group name="groupB">
    <subcycle loop="1">
      <scheme>interstitial_scheme_3</scheme>
      <scheme>primary_scheme_2</scheme>
    </subcycle>
    <subcycle loop="3">
      <scheme>interstitial_scheme_4</scheme>
      <scheme>primary_scheme_3</scheme>
      <scheme>another_interstitial_scheme</scheme>
    </subcycle>
  </group>
</suite>
```

## 3.6 Suite Definition File

The purpose of a CCPP SDF is to describe which physical parameterizations are called in which order for a given model run. SDFs are written in XML format and contain one or more groups of schemes. Within each group, subsets of schemes can be called more than once using subcycles, described earlier in Section 2. Listing 5 shows an example of a simple SDF that contains two groups, `groupA` and `groupB`. In `groupB`, the first set of schemes is called once per timestep, whereas the second set is called three times in a loop. While in reality the names of the schemes are more descriptive (e. g., `gfdl_microphysics` or `mynn_pbl`), the notation in Listing 5 hints at a fundamental difference between traditional physics packages and the CCPP. In the former, the physics driver plays a central role in connecting the various physical parameterizations and presenting them to the host model as one entity. These physics drivers are written by hand and are often many thousand lines long. They contain a large amount of glue code, which we refer to as *interstitial* code, that prepares, converts or transforms variables, computes diagnostics, etc. In the case of CCPP, the driver is a simple host model cap that contains a handful of calls to a standardized, auto-generated CCPP API, which in turn calls the auto-generated physics caps. While the CCPP Framework





is capable of performing certain automatic conversions, such as unit conversions, the majority of the code that in traditional

physics packages resides in the physics drivers must be placed in CCPP schemes. These schemes are referred to as interstitial schemes, as opposed to primary schemes (microphysics, planetary boundary layer, ...). Both types of schemes are written and used in the same way. Future functionality added to the CCPP framework may remove the need for interstitial schemes as the framework matures. Minimization of such schemes is desirable for simplicity and true interoperability of physical parameterizations, since they are often tied to a specific host (i. e., "leftover" glue code from a host's previous hand-coded

physics driver).

### 3.7 Code generator

The heart of the CCPP Framework is the code generator, which, at build time, auto-generates the CCPP API and the physics caps from the metadata. The code generator consists of Python scripts, with `ccpp_prebuild.py` being the main program that gets called by the host model before or as part of the build process. The code generator expects one mandatory argument, a

355 configuration file `ccpp_prebuild_config.py`. This file is host-model dependent, resides with the host model code, and contains information on where to find physics schemes (i. e., depending on where the CCPP physics repo is located within a host's directory structure), kind, type, and variable definitions, etc. The code generator can produce caps for one or more CCPP physics suites simultaneously if a list of suites (which is translated internally into a list of suite definition filenames) is passed as a command-line argument to `ccpp_prebuild.py` (if omitted, all available suites are used).

In addition to the API and the caps, the code generator also produces information for the build system in the form of CMake and GNU Make include files or shell scripts. These contain lists of files that the host model build system must compile to execute the physical parameterizations in the specified suites: kind, type, and variable definitions, physics schemes, and all auto-generated files. The code generator further produces two files to aid development efforts: an HTML document containing a list of all variables defined by the host model, and a LATEX document containing a list of variables used by any of the physics

schemes in any of the selected suites. Figure 4 contains a flowchart of the `ccpp_prebuild.py` code generator with its inputs and outputs.

The auto-generated caps contain the minimum code necessary to execute the physics schemes in each specified SDF. It includes calling the physical parameterizations as well as the various variable manipulations mentioned earlier, namely automatic unit conversions and deblocking of blocked data structures. These transformations may be skipped, depending on

whether a variable is active or on the variable intent (see Section 3.5.3). In addition, to aid development and debugging efforts, the framework can insert tests for variable allocations, size and dimensionality into the auto-generated caps. This feature is activated by passing `-debug` to `ccpp_prebuild.py`. Depending on the host model build system and the configuration in `ccpp_prebuild_config.py`, the main program `ccpp_prebuild.py` is called from a specific directory, in-source or out-of-source, in the build tree. The command-line argument `-verbose` can be used to increase the logging output from the

code generator.

Two additional utilities are included in the CCPP Framework: The script `metadata2html.py` converts scheme metadata, inline documentation in Doxygen format, and additional documentation into a complete scientific documentation for CCPP





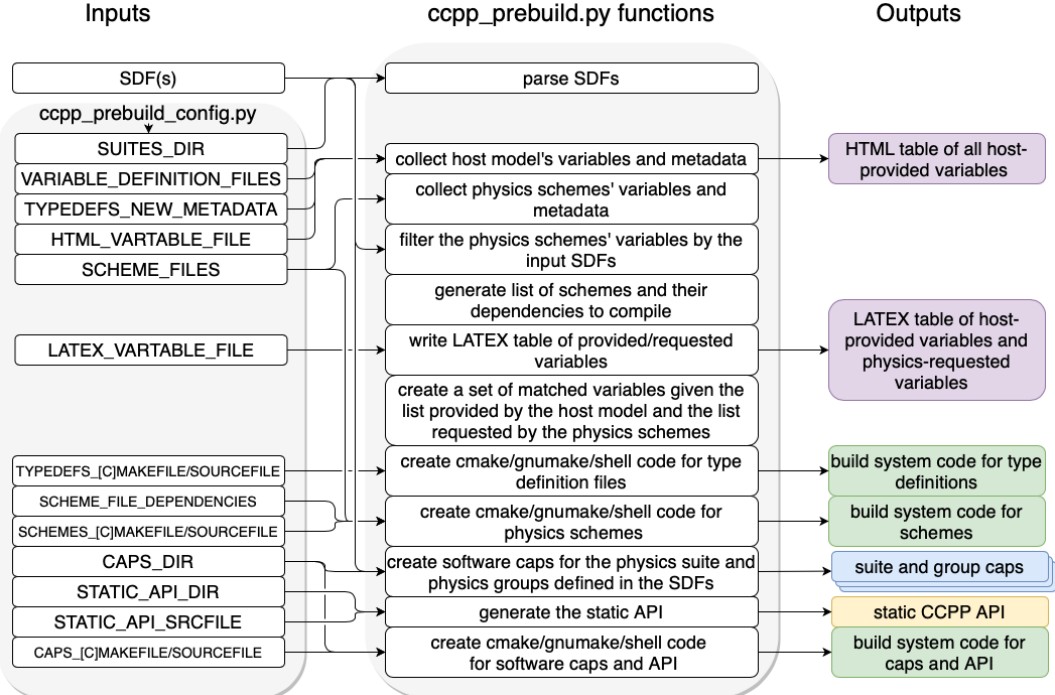

**Figure 4.** Flowchart of the CCPP Framework code generator `ccpp_prebuild.py` with its inputs and outputs.

Physics. This script is generally called from the same location as `ccpp_prebuild.py` and can either be run in batch mode to process all schemes, or individually for each scheme. For an example, see the scientific documentation for CCPP v6 (Zhang et al., 2022).

Lastly, the script `ccpp_track_variables.py` tracks the flow of a variable through a CCPP suite. This utility is useful for understanding a variable's use within a suite and aims at reducing development and debugging efforts. The CCPP technical documentation (Bernardet et al., 2022) provides more information on its use.

### 3.8 Host model integration

The basic integration of CCPP in a host model is shown in Figure 1. A host model cap is used to separate the entry points to the CCPP Physics and the error handling from the other host model logic. The entry points to the CCPP Physics are defined in the auto-generated CCPP API and are Fortran subroutines with a simple, standardized interface:

```
ccpp_physics_PHASE(ccpp_data, suite_name=..., group_name=..., ierr=...)
```

The first argument, `ccpp_data`, is of type `ccpp_t`, which is provided by the CCPP Framework in module `ccpp_types`. This DDT is used to pass information on block number, thread number, and subcycling to the auto-generated caps. It also contains the CCPP error handling variables in the form of an error code and an error message. The `ccpp_data` variable is typically a module variable in the host model cap. For more complicated models that use blocked data structures



and OpenMP threading to process blocks of data in parallel in the run phase, there can be multiple `ccpp_data` variables (see Section 4.2). The host model is responsible for initializing `ccpp_data` before initializing the CCPP physics using

`ccpp_physics_init`, and for setting the block and thread number before any of the calls into the CCPP API. Upon returning from the physics, the host model must handle any errors in the physics by inspecting the `ierr` variable. For example, a value other than zero indicates an error and the host model can print the error flag and error message from `ccpp_data` before gracefully stopping the model. The `suite_name` argument to `ccpp_physics_PHASE` is mandatory and switches between one or more suites that were compiled into the executable. The `group_name` argument is optional and can be used

to call the physics phase `PHASE` for a particular group only. If omitted, phase `PHASE` is executed for all groups in the order specified in the SDF. As mentioned earlier, subcycling of schemes when so defined in the SDF (`<subcycle loop="N">` with N>1) only happens in the run phase, for all other phases `N` is set to 1. Examples for simple and more complex host model caps are described in the following Section 4.

## 4 Examples of use

### 4.1 CCPP Single Column Model

The CCPP SCM simulates the time evolution of the state of the atmosphere over a one-dimensional vertical column extending from the Earth's surface upwards. Lower boundary conditions, such as fluxes of heat and moisture, are obtained from pregenerated datasets or from the land surface, ocean, and sea ice parameterizations. Lateral boundary conditions are obtained from forcing datasets originating from field campaigns, three-dimensional models, or a combination of both. The feature that

distinguishes the CCPP SCM from other SCMs is that it is CCPP-compliant, that is, it contains a CCPP host model cap that allows it to be used with the CCPP Physics.

The integration of CCPP in the SCM is simple and can serve as a template for implementations in other host models. Listing 6 in the Appendix contains a simplified version of the CCPP implemented in the SCM. In this basic setup, all data is stored contiguously and no threading is used. In each CCPP phase, the physics are called for all groups in the order listed in

the SDF.

The CCPP SCM v6 supports six physical suites (that is, six SDFs) that invoke a total of 23 physical schemes from the CCPP Physics. These suites were assembled to support hierarchical testing and development to support research and forecast requirements in short- and medium-range weather.

Figure 5 shows an example for SCM runs using two different physics suites for the 2020 Low Summertime CAPE case from

420 the UFS Case Studies Catalog (Sun et al., 2021). The figure also illustrates the CCPP capability to output diagnostic tendencies for the contributing physical processes, a critical tool to develop and improve the physical parameterizations in the CCPP.



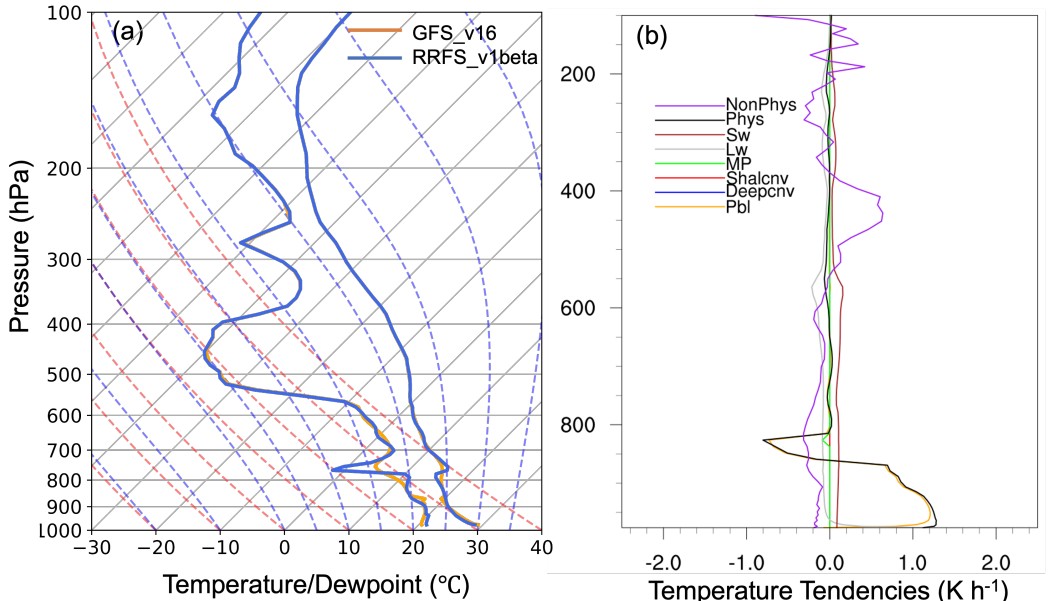

**Figure 5.** (a) Skew-T Log-P plot for SCM runs with physics suites GFS v16 (GFS_v16) and RRFS v1beta (RRFS_v1beta), initialized at 00 UTC on July 23, 2020, at the Atmospheric Radiation Measurement Southern Great Plain central facility site (36.61degN, 97.49degW) and valid at forecast hour 15. (b) Tendencies from the GFS v16 physics suite valid at the same time for the different contributing processes (Pbl: planetary boundary layer, Deepcnv/Shalcnv: deep/shallow convection, MP: microphysics, Lw/Sw: longwave/shortwave radiation, Phys: all physics tendencies, NonPhys: non-physics tendencies, e. g., dycore/advection).

## 4.2 Unified Forecast System

The UFS is an example of a complex Earth system model that calls CCPP from the atmospheric model through the host model cap and optionally from within the Finite-Volume Cubed-Sphere (FV3) dynamical core. Since in the UFS the dynamics
timestep is shorter than the general physics timestep, we refer to the physics called from the dynamical core as tightly coupled or fast physics, as opposed to the traditional or slow physics that are called through the atmospheric host model. The fast physics currently consist of a single scheme, the saturation adjustment step for the GFDL cloud microphysics (Zhou et al., 2019). The UFS uses the capability to call individual groups of physics from the SDF to implement the slow and fast physics and perform other operations between the calls to radiation, stochastic physics and the remaining slow physics. The schematic
diagram of the UFS in Figure 6 illustrates this complexity. The UFS also uses blocked data structures to execute the time integration (run) phase in parallel using multiple OpenMP threads. Listing 7 in the Appendix contains an abstraction of the host model cap in the UFS, called `CCPP_driver.F90`, which is notably more complicated than the simple host model cap for the CCPP SCM (Listing 6).

Recently, the capability to call aerosol chemistry parameterizations as part of the physics was added to the UFS (Barnes et al.,
2022). The CCPP Framework has also been connected to the Community Mediator for Earth Prediction System (CMEPS; Earth



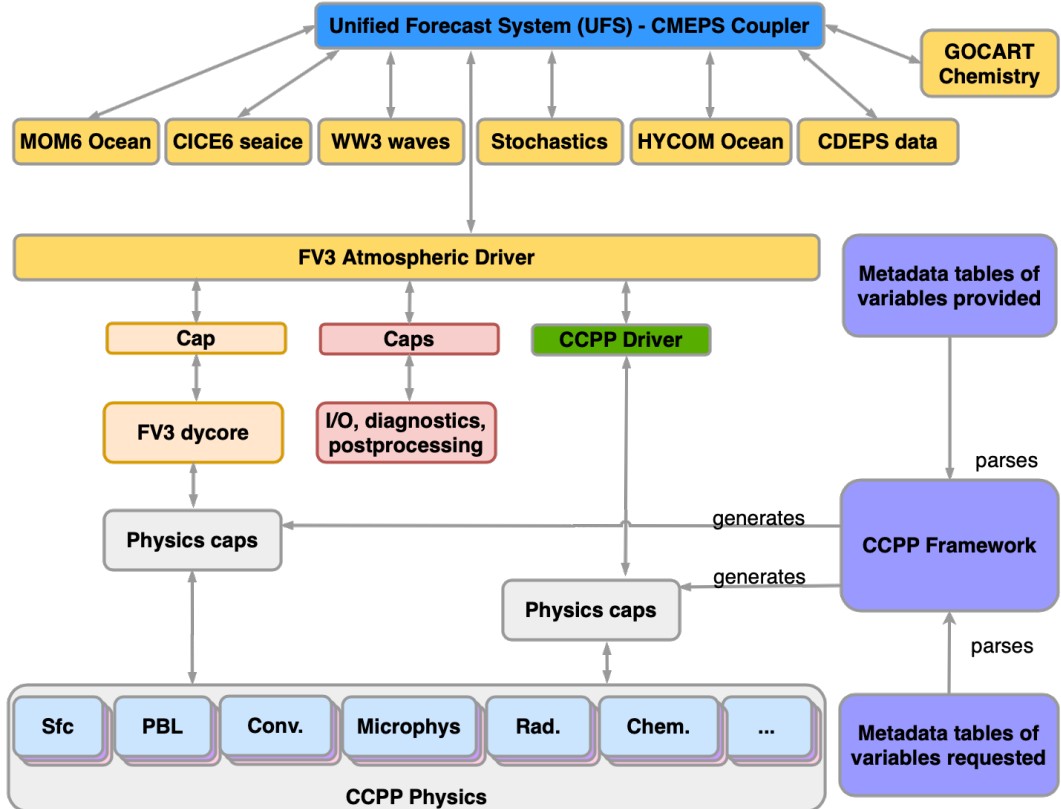

**Figure 6.** Schematic representation of the integration of CCPP in the UFS.

System Community Modeling Portal, 2022b) to enable the computation of thermodynamic fluxes between the atmosphere and the underlying ocean and sea ice using a predefined grid to exchange information among the various component models (U. Turuncoglu, NCAR, priv. comm.). Both developments were recently merged into the official code repository and are omitted in Figure 6.

The CCPP has been included as a subcomponent of all UFS public releases thus far, with the most recent one being the UFS SRW App v2 (UFS Community, 2022). The SRW App is a UFS configuration intended for research and operational applications on temporal scales of hours to a few days. The SRW App v2 supports simulations with four physics suites, which invoke a total of 21 CCPP schemes. One of the suites is a baseline configuration representing the physics used in the currently operational GFS v16, while the other three suites represent physics permutations with potential for inclusion in the upcoming

operational implementation of the RRFS v1beta, a multi-physics convection-allowing ensemble, at NOAA. To illustrate the use of CCPP in the UFS SRW App over the contiguous United States, Figure 7 shows precipitation forecasts produced with the GFS v16 suite (which invokes a microphysics parameterization with saturation adjustment called from the dynamical core) and the RRFS v1beta suite, which involves a different microphysics scheme that does not have a tightly coupled component in the dynamical core. The two forecasts have notable similarities, as both indicate severe thunderstorms with associated damaging



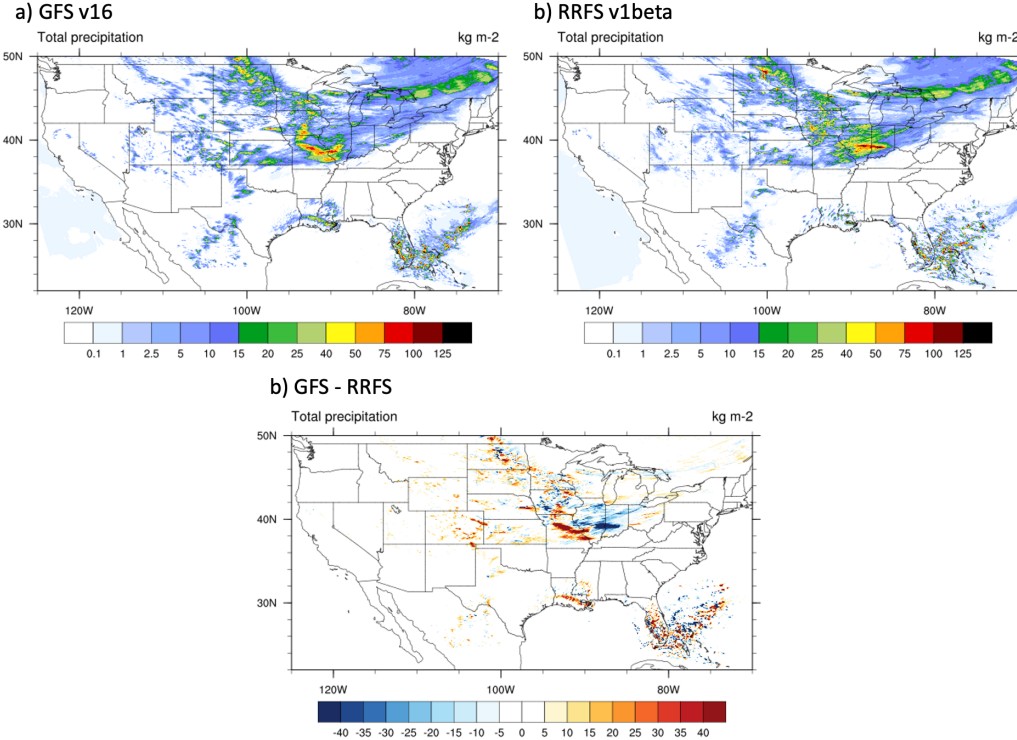

**Figure 7.** UFS SRW App v2 3-km forecasts of 24-h accumulated precipitation ($\mathrm{kg\,m^{-2}} = \mathrm{mm}$) initialized at 00 UTC on June 15, 2019, and valid at 00 UTC on June 16, 2019, created with the a) GFS v16 suite and b) RRFS v1beta suite. The difference between the two forecasts is shown in panel c).

winds, large hail, and heavy rainfall developing across portions of Iowa, Missouri, and Illinois in the evening of June 15. However, the location of heavy rainfall differs between the forecasts. The ability of the CCPP-enabled UFS to readily use multiple suites to create diverse forecasts enables experimentation for investigative research and for ensemble forecasting.

## 5   Discussion and Outlook

The CCPP Framework and Physics are designed to increase the interoperability of physical parameterizations and host mod-
eling systems. The system relies on standardization, metadata information, clear documentation, and constraints on the format of the Fortran source code to automatically generate the interfaces that connect the parameterizations and the host model. The simple yet clear metadata standard builds on established efforts such as the CF metadata conventions and the UDUNITS package. The need to define metadata, follow coding standards and provide clear documentation are additional efforts that developers of parameterizations and host models have to make compared to traditional, hard-coded physics drivers. However,
the ability to add new parameterizations, transfer them between models and connect them with other parameterizations without





having to rewrite the code or make assumptions on units or dimensions outweighs the costs. The CCPP is also computationally performant: While traditional physics drivers contain a lot of code branching to switch between different parameterizations of one type, the auto-generated physics suite caps benefit from knowing at compile time which parameterizations are called and in which order, and therefore require no code branching.

Since its inception in 2017, the CCPP has been integrated into several modeling systems: CCPP-SCM, UFS, the Navy's next-generation NEPTUNE model, NCAR's MUSICA infrastructure for chemistry and aerosols, and recently also in experimental mode in NASA's Goddard Earth Observing System (GEOS; B. Putman, NASA, priv. comm.). The CCPP Physics has received contributions from the wider community and contains more than 30 parameterizations of physical processes. A multi-institutional CCPP Physics code management committee has been established to provide governance and guidance on

developing, testing and integrating of new parameterizations. The CCPP Framework development, which until now has been concentrated on a small group of developers from DTC with input from several NCAR and NOAA laboratories, is being broadened. To further enhance the capabilities of the CCPP Framework and enable its use in additional modeling systems, NOAA and NCAR in 2019 agreed on a joint development effort as part of a wider Memorandum of Agreement on common infrastructure elements of Earth modeling systems (UCAR-NCAR et al., 2019). As part of this effort, work is underway to make

the NCAR flagship modeling systems Community Atmosphere Model (CAM; a component of the Community Earth System Model) and MPAS (Model for Prediction Across Scales) CCPP-compliant and to extend the CCPP Framework code generator to satisfy the needs of the growing number of systems it will support.

While the CCPP has reached maturity and is on track to transition to operational implementations of the UFS at NOAA in 2023, it is crucial to look ahead and identify developments needed to meet future requirements. An important aspect of the

future development of the CCPP is leveraging emerging technologies, such as using hardware accelerators (GPU, FPGA, etc.) and modern processor vectorization capabilities (see Zhang et al., 2020, for an example). As physical parameterizations are augmented to run efficiently on such systems, the CCPP Framework must support these developments in the auto-generated caps by reducing the data movement and transforming the data to benefit from the increased parallelism and vector processing. Scientific innovations, such as machine learning and the move towards three-dimensional physical parameterizations, also need

to be considered in future development. Machine learning and artificial intelligence are promising candidates for improving the accuracy of parameterizations and/or significantly reducing the computational costs (see Irrgang et al., 2021, for a review). These technologies make heavy use of Python libraries such as scikit-learn, which aligns well with the fact that the CCPP Framework code generator is written entirely in Python and relies on clearly defined metadata and standardized interfaces. The use of Python in the CCPP Framework is also compatible with efforts to provide simpler, domain-specific interfaces to the

complex codes used in Earth system models (McGibbon et al., 2021).

Unlike the CCPP Physics, the CCPP Framework to date has received few contributions from the community. This is expected to change as CCPP is implemented in more host modeling systems for research and operations, and has additional public releases. Because the Framework automatically generates the interfaces between the physical parameterizations and the host model, is written in Python, and builds on a clearly defined metadata standard, there are ample opportunities for improvements

and new capabilities. Examples are tools that depict the flow of a variable through a physics suite (recently developed at DTC),



improved diagnostic capabilities, automated saving of restart information, and many more. For both the Framework and the Physics, the CCPP developers welcome contributions from the community via the open source development repository hosted on GitHub (https://github.com/NCAR/ccpp-framework and https://github.com/NCAR/ccpp-physics).

*Code availability.* The Common Community Physics Package (CCPP) Single Column Model v6.0.0 (with Physics and Framework) (DOI: 10.5281/zenodo.6896438) are available for download as a single `.tar.gz` file at https://zenodo.org/record/6896438#.Yus5MJPMITs. The associated technical documentation (DOI: 10.5281/zenodo.6780447 ) is available for download at https://zenodo.org/record/6780447# .Yus6IpPMITs. The releases are also available for download from the the NCAR GitHub repositories `ccpp-scm`, `ccpp-framework`, `ccpp-physics`, `ccpp-doc` as tags `v6.0.0`. A recursive clone of the CCPP SCM repository (`git clone -b v6.0.0 -recursive https://github.com/ncar/ccpp-scm`) will download the SCM, the Framework and the Physics. The technical documentation must be cloned separately (`git clone -b v6.0.0 https://github.com/ncar/ccpp-doc`). All downloads, documentation and supporting information are also available at https://dtcenter.org/community-code/common-community-physics-package-ccpp.

Copyright 2022, NOAA, UCAR/NCAR, CU/CIRES, CSU/CIRA. Licensed under the Apache License, Version 2.0 (the "License"). You may obtain a copy of the License at http://www.apache.org/licenses/LICENSE-2.0. Unless required by applicable law or agreed to in writing, software distributed under the License is distributed on an "AS IS" BASIS, WITHOUT WARRANTIES OR CONDITIONS OF ANY KIND, either express or implied. See the License for the specific language governing permissions and limitations under the License.



## Appendix A: Examples of use – CCPP host model caps

## A1    CCPP Single Column Model

**Listing 6:** Simplified version of `scm/src/scm.F90`.

```fortran
module scm_main

   implicit none

contains

   subroutine scm_main_sub()
      use :: ccpp_types, only : ccpp_t
      use :: ccpp_static_api,                   &
            only: ccpp_physics_init,            &
                  ccpp_physics_timestep_init,   &
                  ccpp_physics_timestep_run,    &
                  ccpp_physics_timestep_finalize, &
                  ccpp_physics_finalize
      character(len=256) :: suite_name
      type(ccpp_t) :: cdata

      ! Initialize the CCPP framework ccpp_data variable
      cdata%blk_no = 1
      cdata%thrd_no = 1

      ! Set suite name when reading namelist, then initialize physics
      call ccpp_physics_init(cdata, suite_name=trim(suite_name), ierr=ierr)

      do i = 1, n_timesteps
         call ccpp_physics_timestep_init(cdata, suite_name=suite_name, ierr=ierr)
         call ccpp_physics_run(cdata, suite_name=suite_name, ierr=ierr)
         call ccpp_physics_timestep_finalize(cdata, suite_name=suite_name, ierr=ierr)
      end do

      call ccpp_physics_finalize(cdata, suite_name=trim(suite_name), ierr=ierr)
   end subroutine scm_main_sub

end module scm_main
```



## A2  Unified Forecast System

**Listing 7:** Simplified version of `FV3/ccpp/driver/CCPP_driver.F90`.

```fortran
module CCPP_driver

  use ccpp_types,     only: ccpp_t

  use ccpp_static_api, only: ccpp_physics_init,              &
                             ccpp_physics_timestep_init,     &
                             ccpp_physics_run,               &
                             ccpp_physics_timestep_finalize, &
                             ccpp_physics_finalize

  use GFS_data,       only: GFS_control

  implicit none

  type(ccpp_t),                              target :: cdata_d
  type(ccpp_t), dimension(:,:), allocatable, target :: cdata_b

  ! ccpp_suite is set during the namelist read by the host model
  character(len=256) :: ccpp_suite
  integer            :: nthreads

  public CCPP_step

contains

  subroutine CCPP_step(step, nblks, ierr)

    character(len=*), intent(in)  :: step
    integer,          intent(in)  :: nblks
    integer,          intent(out) :: ierr
    ! Local variables
    integer :: nb, nt
    integer :: ierr2

    ierr = 0
    ...
```





**Listing 7 (continued).**

```fortran
...
if (trim(step)=="init") then

  ! Get and set number of OpenMP threads (module
  ! variable) that are available to run physics
  nthreads = omp_get_max_threads()

  ! For physics running over the entire domain,
  ! block and thread number are not used
  cdata_d%blk_no = 1
  cdata_d%thrd_no = 1

  ! Allocate cdata structures for blocks and threads
  allocate(cdata_b(1:nblks,1:nthreads))

  ! Assign the correct block and thread numbers
  do nt=1,nthreads
    do nb=1,nblks
      cdata_b(nb,nt)%blk_no = nb
      cdata_b(nb,nt)%thrd_no = nt
    end do
  end do

else if (trim(step)=="physics_init") then

  ! Since the physics init step is independent
  ! of the blocking structure, use cdata_d.
  ! Since we don't use threading in the host,
  ! we can allow threading inside the physics.
  GFS_control%nthreads = nthreads
  call ccpp_physics_init(cdata_d, suite_name=trim(ccpp_suite), ierr=ierr)

else if (trim(step)=="timestep_init") then

  GFS_control%nthreads = nthreads
  call ccpp_physics_timestep_init(cdata_d, suite_name=trim(ccpp_suite), ierr=ierr)
...
```



**Listing 7 (continued).**

```fortran
    ...
    else if (trim(step)=="radiation" .or. &
             trim(step)=="physics" .or.   &
             trim(step)=="stochastics") then

       ! Set number of threads available to physics schemes
       ! to one, because threads are used for blocking
       GFS_control%nthreads = 1

!$OMP parallel num_threads (nthreads) reduction (+:ierr)
       nt = omp_get_thread_num()+1
!$OMP do schedule (dynamic,1)
       do nb = 1,nblks
         call ccpp_physics_run(cdata_b(nb,nt), suite_name=trim(ccpp_suite), &
                               group_name=trim(step), ierr=ierr2)
         ierr = ierr + ierr2
       end do
!$OMP end do
!$OMP end parallel

    else if (trim(step)=="timestep_finalize") then

       GFS_control%nthreads = nthreads
       call ccpp_physics_timestep_finalize(cdata_d, suite_name=trim(ccpp_suite), ierr=ierr)

    else if (trim(step)=="physics_finalize") then

       GFS_control%nthreads = nthreads
       call ccpp_physics_finalize(cdata_d, suite_name=trim(ccpp_suite), ierr=ierr)

    else if (trim(step)=="finalize") then

       deallocate(cdata_b)

    end if

  end subroutine CCPP_step

end module CCPP_driver
```



*Author contributions.* Dr. Dominikus Heinzeller (DH) developed the CCPP Framework, ported physical parameterizations to CCPP Physics,
integrated CCPP in the UFS, assisted the Naval Research Lab (NRL) with the integration of CCPP in NEPTUNE, and lead the CCPP code
management until the end of 2021. Dr. Ligia Bernardet (LB) led the CCPP project, performed project management duties, and oversaw
the development of the CCPP Framework, Physics, and associated documentation. Dr. Grant Firl (GF) assisted with the CCPP Framework
development, led the technical development of the CCPP Physics library, integrated CCPP in the SCM, and led the CCPP code management
since 2022. Dr. Man Zhang (MZ) ported physical parameterizations to CCPP Physics and was the main developer of the CCPP scientific
documentation. Dr. Xia Sun (XS) contributed to the testing of CCPP within the UFS Case Studies project, and Dr. Mike Ek (ME) acted as
co-lead for the CCPP project. DH and LB created this manuscript, GF, MZ and XS contributed to the graphics. All authors reviewed the
manuscript.

*Competing interests.* No competing interests are present.

*Acknowledgements.* The authors would like to acknowledge the invaluable contributions made by Timothy Brown (then of CU/CIRES at
525 NOAA/GSL) and David Gill (then at NCAR) to early versions of the CCPP. Further acknowledged are Laurie Carson (then at NCAR),
Julie Schramm (then at NCAR), Samual Trahan (CU/CIRES at NOAA/GSL), Mike Kavulich (NCAR) and Weiwei Li (NCAR) for their
contributions to the CCPP physics and documentation. The authors would also like to thank Steve Goldhaber (NCAR), Cheryl Craig (NCAR)
and Francis Vitt (NCAR) for their valuable input into the CCPP Framework development, and Duane Rosenberg (NOAA/GSL) for reviewing
this manuscript. Lastly, the developers at NOAA EMC, NOAA GSL, NOAA GFDL and the Naval Research Lab are acknowledged for their
constructive feedback that led to a better design of the CCPP. This work is supported by the UFS Research-to-Operation (UFS-R2O) project,
sponsored by the National Weather Service Office of Science and Technology Integration Modeling Program Divison and the NOAA Oceanic
and Atmospheric Research Weather Program Office.



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
