# Peer review of "The Common Community Physics Package (CCPP) Framework v6"

_EGUsphere, 2022_

## Referee Comment (RC1)

General Comments:

This paper aims to introduce the CCPP framework that is for flexibly using atmospheric physical parameterization schemes in various model simulations. I think it deserves attentions from model developers, model users and framework developers, as it presents new thoughts for integrating and using parameterization schemes, and the CCPP framework has already been used in real model development. However, I think the current version of this paper should be significantly improved after revisions. The current context seems hard to understand, as it focuses on the design and implementation of the CCPP framework, without sufficient motivations and analysis that are important to make this paper more attractive and understandable.

Specific comments:

1. About motivations of developing the CCPP framework.

It will be welcome to give examples for motivating this framework, e.g., the current status of physical packages, real scientific requirements that the current physical packages cannot serve but the CCPP framework can.

2. About motivations of the design and implementation of the CCPP framework.

It seems that the main target of the CCPP framework is to enable users to flexibly group physical schemes into a package via an XML configuration file (Listing 5). To achieve such a target, authors prefer a solution that automatically generates Fortran code of callers corresponding to the configuration file. A challenge here is how to achieve automatic caller code generation, as the generated caller code should match the argument list of the corresponding callee that is a physical scheme. To overcome this challenge, authors propose to describe each argument of a physical scheme via the corresponding metadata (listing 2), including the variable name, dimension order, data type, in/out attribute of each argument. Moreover, authors seem to still keep the rule that a caller and a callee share the same data structures and the same variable name space of all arguments, and parse argument values between each other via shared memory space. Under such rule, a kind of metadata is designed for generating the Fortran code of declaring all model variables that are shared by all physical schemes.

Under the above understanding (I am sorry if I misunderstand), I think this paper should state why the current technical solution of the CCPP framework is almost the best option. I am a little afraid that the current solution that highly depends on automatic code generation and the development metadata may introduce new challenges to users, e.g., work for studying the metadata rules and developing metadata files, more phases in compiling the codes, and efforts for guaranteeing the consistency between the metadata files and codes.

There may be other possible solutions. For example, there can be native Fortran code files for declaring all variables and data types used in the physical package, a native Fortran code file that enumerates all callers each of which corresponds to physical scheme, and a native Fortran

code file for controlling the usage flow of a set of physical schemes under the XML configuration file. Even when a Fortran code file is not flexible enough for achieving such a control flow, C/C++ can work cooperatively with simple Fortran callees with a few arguments (a complex Fortran callee can be enclosed in a simple callee that parses arguments to the complex callee via global data). Such a simple design may not achieve all goals of the CCPP framework. However, I believe that comparisons with simple designs can make this paper more understandable.

3. About "Neither Fortran common blocks nor the local definition or importing of physical constants or functions from other physical parameterizations are allowed. Physical constants must be passed to the physics schemes via the argument list. Each CCPP scheme may define its own dependencies (i. e., Fortran, C, C++, . . . modules) that are required for the scheme, as long as these do not depend on the presence of other schemes and vice versa." and "To ensure a consistent set of physical constants for use by all physical parameterizations, these constants must be defined by the host model and passed to the physics via the argument list, in other words, they are treated like normal variables. "

   I cannot fully understand the necessity of the above restriction rules. I tend to guess that a scheme should not share any global data (including constants) with other schemes and the model, and all arguments of a scheme should be passed via the argument list. However, the model and different schemes essentially share the same data structures and the same variable name space of all arguments. Moreover, it seems difficult for the CCPP framework to detect the case of sharing global constants as well as global variables among different schemes.

4. About Figure 2 and Listing 1.

   Can authors give an example about the corresponding motivation? When the blocked data structure is initialized from a round-robin parallel decomposition and a block contains multiple columns, a contiguous array generated by the code in Listing 1 does not match the horizontal grid. As each scheme share the parallel decomposition of the model, is there any restriction about the parallel decomposition?

5. About dynamically linked library (DLL).

   I note that an old version of CCPP used DLLs to enclose schemes. Does the new version still use DLL technique? If not, it will be welcome to introduce the corresponding reason in revisions.

6. Can users use CCPP in a recursive manner, e.g., a scheme works as the host model of other schemes? If cannot, does CCPP has the functionality to forbidden the recursive usage?

7. Can CCPP be used in an incremental manner, e.g., a part of schemes are used with CCPP while a part of schemes are called in a traditional manner? If can, is there any restriction?

8. About sharing the same schemes among different models.

It will be welcome to discuss about this question. Given that a set of schemes have been adapted to CCPP and used in model A, what efforts should be made for using these schemes with CCPP in model B that has the variable names (as well as the dimension order) different from the model A?

---

## Author Comment (AC1)

**egusphere-2022-855 - final response**

Table of contents:

**Answers to Editor**

The authors thank the editor for reminding us of the rules and restrictions about colored text in the manuscript, as well as about checking our Figures 5 and 7 using the Coblis – Color Blindness Simulator.

1. We removed the syntax highlighting (i.e.colors) from all listings.
2. We checked both figures using the color blindness simulator (and visually, as the first author is himself color blind). We made revisions to Figure 5, both panels, to aid readability, but concluded that no changes were needed for Figure 7. The colors in Figure 7 have no actual meaning, they only separate the building blocks of an earth system model into categories. Even when selecting the most extreme monochromatic filters, the building blocks are still distinguishable.

**Answers to Reviewer 1**

The authors thank the reviewer for their careful and detailed review of our manuscript. We have provided answers to their eight comments or questions below and partially also in the revised manuscript (see below for details).

1. **About the motivations for developing the CCPP Framework.**

   Many state-of-the-art NWP models, such as the NOAA operational Global Forecast System (GFS), the NCAR Weather Research and Forecast (WRF), and the NCAR Model for Prediction Across Scales (MPAS) rely on a *physics driver* to call the parameterizations. These drivers use conditional statements to make decisions about which primary parameterizations should be invoked at runtime based on user-specified options. The problem with these drivers is that they comprise hundreds, if not thousands, lines of code because of all the variables that need to be pre- and post-processed around the parameterizations. This approach compromises computational extensibility (because adding new parameterizations means making the driver even longer), computational efficiency (because compiler optimization is hindered by all the possible runtime permutations) and ease-of-use (because code inspection is hindered by the many paths and branches on the code). Since the host models that use CCPP do not need a *physics driver*, they avoid these shortcomings.

   Another problem with models having their own physics driver and parameterizations is that it is not straightforward for users and developers to unambiguously identify the quantities that are passed in and/or out of the parameterizations. Through the use of metadata to describe these quantities, along with the requirement that variables be passed via the argument list, CCPP-compliant host models and parameterizations expose these quantities, leading to transparency and ease-of-use. Additionally, the use of metadata for the variables communicated between host model and parameterizations, along with adherence to the modified Kalnay rules (Doyle et al., 2020), foster interoperability. This opens the door to the previously-absent ability to streamline parameterization development for use in multiple host models, reducing the need for developers to maintain separate code bases for different host models.

   To provide additional motivation for the development of the CCPP Framework, the following text was added to Section 1 (Introduction):

   *"Many current NWP models, such as the GFS and the NCAR Model for Prediction Across Scales (MPAS), rely on physics drivers to to call parameterizations embedded in the host model code base. This lack of separation of concerns between host and physics compromises computational extensibility (because adding new parameterizations makes long drivers even longer), computational performance (because compiler optimization is hindered by all the possible runtime permutations), and ease-of-use (because code inspection is made difficult by the many paths and branches on the code). Additionally,*

*this approach does not empower parameterization developers to simplify the
development workflow by using the same code base in various host models."*

2. **About the motivations for the design and implementation of the CCPP Framework.**

The reviewer is correct with the assessment stated in the first paragraph of their
Comment 2, including the section about the shared memory space between the host
model and the CCPP physics. The present manuscript is a presentation of the CCPP
Framework, which falls into the category of "Model description papers" in GMD
([https://www.geoscientific-model-development.net/about/manuscript_types.html#item](https://www.geoscientific-model-development.net/about/manuscript_types.html#item)),
and not a discussion or evaluation of different possible methods. We will therefore keep
our answers to this question mostly limited to this document.

It is true that alternatives exist to the approach chosen, each with its pros and cons. The
design of the CCPP Framework has undergone several changes and optimizations
throughout its development phase until it reached the current state as presented in the
paper. The current CCPP Framework consists of a Python code generator that relies on
metadata supplied in addition to the (host model or CCPP physics) code. The code
generator produces static Fortran interfaces that connect the host model to the physics
in a preprocessing step before the code gets compiled, for one or more CCPP suites
(combinations of CCPP-compliant physical parameterizations) at a time. The
auto-generated code gets compiled together with the host model code to produce a
static executable (at least w.r.t. CCPP Framework and Physics).

The initial version of the CCPP Framework used a different approach, as the reviewer
also referred to in his Comment 5: Compliant physics schemes were compiled into a
dynamic library that was linked to the executable at compile time. This solution offered
complete runtime control over which physics schemes to call. This is in contrast to the
final approach taken that requires specifying one or more suites at compile time, making
them available to choose at runtime. The main disadvantages of the dynamic/dll
approach were that the host model (usually in Fortran) was calling C functions with
arguments (names of schemes to run), which the C functions would convert into `dlload`
commands from the dynamic library. As a result, arguments were passed by reference
(location in memory) from Fortran to C space to Fortran space. Besides the poor
computational performance of this approach, it led to the inability to infer from reading
the code what actually happens at runtime (something that model developers, especially
Fortran developers are highly used to and require), and valuable tools like debuggers or
compiler checks (especially debug flags) couldn't be used. Further, operational
requirements from NOAA did (and still do) require static executables for portability and
debugging of crashes in operations. Many operational centers also use Cray machines
as their production or research high performance computing systems, and Cray
traditionally prefers static linking over dynamic linking. The current design of the CCPP
Framework enables better compiler optimizations, compiler checks, debugging, "reading"
the final code that gets compiled into the executable, and static linking. It also delivers

the necessary performance for operations and presents a compromise between flexibility and performance by making multiple suites available at runtime.

The reviewer also mentions alternative approaches such as native Fortran code files, and describes the need to create consistent metadata as a drawback of the current solution. It is correct that writing metadata and keeping it consistent with the Fortran code is an additional step, as is the need to parse it with Python and generate static interfaces prior to compiling the model. However, the benefits of this approach are ample: The CCPP developers have full control over the metadata standard and format and can adjust or extend it as required. Python is also incredibly powerful in parsing these metadata files and generating code or other useful information from it. It is much more difficult to encode and parse arbitrary metadata in Fortran (or C, C++). The Python code generator, as well as the static compilation, are able to catch nearly all possible inconsistencies between the metadata and the Fortran code. It further gives us the opportunity to perform automatic unit conversions (implemented), array transformations (planned), dimension/out-of-bounds/allocation checks (implemented), and it allows us to generate diagnostic tools such as visualizations of how a given variable traverses through a suite (recently developed), to name a few. Last but not least, one of the most powerful yet not utilized features of the current CCPP Framework is that it is language-agnostic and can, after adding the necessary templates to the Python code generator, produce interfaces between host model and physics code in another language (e.g. C), or interfaces between mixed languages (e.g. C host model calling Fortran physics), and it can be augmented to add preprocessor directives (e.g. OpenMP/OpenACC pragmas) to the auto-generated code.

We added the following information to the manuscript at the end of Section 2 (Design) to provide a bit more background to the reader:

*"… However, this was considered as an advantage of the current CCPP Framework over alternative approaches that rely on native Fortran code files for declaring variables and types.*

*It should be noted that an early version of the CCPP Framework used a different approach. All available compliant physics schemes were compiled into a dynamic library that was linked to the executable at build time, without specifying an SDF. The SDF was parsed at runtime, and the schemes were called in the order specified in the SDF using a Fortran-C-Fortran interface, in which the C layer dynamically loaded (`dlload`) the scheme from the Fortran physics library. This dynamic approach had many disadvantages, for example that arguments were passed by reference from Fortran to C to Fortran space. Besides the poor computational performance, this approach made it much more difficult to debug the code, since developers were not able to "read" the code that gets executed at runtime, and since important tools like debuggers were not able to detect errors. Further, NOAA operations as of today still require static executables for portability and debugging of crashes in production. The current design of the CCPP*

*Framework enables compiler optimizations, compiler checks, debugging, "reading"' the final code that gets compiled into the executable, static linking, delivers the necessary performance for operations, and presents a compromise between flexibility and performance by generating multiple suites at the same time."*

3. **About Fortran common blocks and sharing of data between schemes.**

   We added the following text to the manuscript in Subsection 3.3 (CCPP-compliant schemes):

   *"The restriction on Fortran* `common` *blocks has several reasons. First, a scheme must not share its data (variables, constants, types) with other schemes, and* `common` *blocks not only allow this, but also make it difficult to detect violations of this requirement. Not sharing data with other schemes is to ensure that each scheme is self-contained, and at the same time guarantee that all schemes use a consistent set of physical constants defined by the host model. Second, the requirement for CCPP schemes to be Fortran modules following modern programming standards essentially forbids using old Fortran 77 constructs such as* `common` *blocks. Third, Fortran* `common` *blocks are inherently dangerous, because they permit the declaration of the same block differently in different procedures and evade proper type checking."*

4. **About Figure 2 and Listing 1.**

   The motivation for the capability to combine blocked data structures into contiguous arrays is given in the original manuscript in Section 3.2 (CCPP Phases), first paragraph:

   *"With exception of the time integration (run) phase, blocked data structures must be combined into contiguous data such that a physics scheme has access to all data that an MPI task owns. The need to have access to all data and the limitations on OpenMP threading described in the previous section are a result of potential file Input/Output (I/O) operations, computations of statistics such as minimum values of a variable, interpolation of lookup table data (e.g., ozone concentration from climatology), and global communication during those phases."*

   We added the following text to the beginning of the next paragraph to provide a short motivation as to why host models may make use of blocked/chunked data structures in the time integration (run) phase.:

   *"For better computational performance, NWP models often make use of parallel execution of the physics using multiple threads, in which each thread operates on a subset of the data owned by an MPI task. The exact implementation is model-dependent and can range from passing different start and end indices of a contiguous array to splitting up contiguous arrays into separate blocks. To support the latter case, …"*

CCPP does not impose any restriction on the parallel decomposition of the model or on the distribution of data into separate blocks. Since the physics in CCPP are by definition one-dimensional, i.e. treat each vertical column independently of its neighbors, it is irrelevant if the contiguous (or chunked) data matches the horizontal grid. CCPP Physics that operate on independent vertical columns are therefore compatible with irregular meshes, as it is the case in the NCAR Model for Prediction Across Scales (MPAS), for example.

5. **About the dynamically linked library (DLL).**

   This comment is addressed in this document (and in the manuscript) as part of Comment 2 above.

6. **Recursive CCPP**

   We have so far not encountered the need for recursion in CCPP schemes, and we cannot think of any use case for it. However, there are no restrictions in the CCPP rules that would forbid such a usage. One could, with the appropriate set up of the build system, create a CCPP scheme that is called by a host model and that in turn calls other CCPP schemes (using its own suite definition file). Under the current rules for variable metadata attributes, this would require creating two metadata tables for the scheme, one for its role as a scheme, and one for its role as a "host model". If the need for such recursion ever arises, one could consider modifying/extending the metadata rules such that separate those tables could be combined into one.

7. **Incremental use of CCPP in a model.**

   Absolutely, yes. We made extensive use of this method when we implemented CCPP in the Unified Forecast System (UFS), which is mentioned in Section 4.2 of the manuscript. Termed "hybrid CCPP" at that time, we added calls to CCPP physics inside the existing physics driver files for schemes that had been ported to CCPP. With more and more schemes becoming available, the Suite Definition File grew longer and the number of schemes called in a traditional way became shorter. What is more, with the appropriate preprocessor (CPP) macros we used this "hybrid CCPP" approach to toggle between the CCPP and non-CCPP versions of one or more schemes to ensure that our CCPP implementation was bit-for-bit identical with the original, physics-driver based model.

8. **About sharing the same schemes among different models.**

   Today there are three host models sharing CCPP-compliant parameterizations (UFS, CCPP SCM, and NEPTUNE) and work is in progress to adopt CCPP as a component of various NCAR models. Therefore, it is demonstrated that CCPP allows sharing parameterizations among various hosts.

When a host model uses the CCPP, it is straightforward to connect it to primary parameterizations used by other CCPP-compliant hosts. The first step is to examine which quantities are passed in and/or out of the parameterization (this information can be found in the metadata associated with that scheme). The next step is to identify which of those quantities are already present in the host model and, if not yet present, create metadata for them. Quantities that are not present in the host model need to be derived from existing variables either in the host model itself or in interstitial schemes called before the primary parameterization. The process to connect a CCPP-compliant host model to new parameterizations is described in Chapter 9 of the CCPP Technical Documentation. This how-to information will not be included in the manuscript to avoid excessive detail.

Sometimes the desired quantity exists in the host model but its array has a different dimension than the parameterization expects. In the future, it is expected that the CCPP Framework will be able to identify these discrepancies and automatically transform existing arrays to the expected dimensions, much like how it currently converts units between what a host provides and what a parameterization expects. Until this capability is implemented, it is necessary to manually write a host model cap or interstitial scheme to transform the necessary arrays before entering and after returning from the physical parameterization. This information is now included in the second paragraph of Section 3.5.4: *Currently, a manually written host model cap or interstitial scheme is required to transform all arrays before entering and after returning from the physical parameterization. Work is underway to implement automatic array transformations in a future version of the CCPP Framework.*

**Answers to Reviewer 2**

The authors thank the reviewer for their careful and detailed review of our manuscript. We have provided answers to their general remarks and three specific comments or questions below and partially also in the revised manuscript (see below for details).

1. **General comment on purpose and scope of the paper.**

   The present manuscript is a presentation of the CCPP Framework and falls into the category of "Model description papers" in GMD (https://www.geoscientific-model-development.net/about/manuscript_types.html#item). GMD characterizes "Model description papers" as follows:

   "*Model description papers are comprehensive descriptions of numerical models which fall within the scope of GMD. The papers should be detailed, complete, rigorous, and accessible to a wide community of geoscientists. In addition to complete models, this type of paper may also describe model components and modules, as well as frameworks and utility tools used to build practical modelling systems, such as coupling frameworks or other software toolboxes with a geoscientific application. The GMD definition of a numerical model is generous, including statistical models, models derived from data (whether model output or observational data), spreadsheet-based models, box models, 1-dimensional models, through to multi-dimension mechanistic models.*"

   GMD further explains that *"The publication should consist of three parts: the main paper, a user manual, and the source code, ideally supported by some summary outputs from test case simulations."* The present manuscript is the main paper, and a comprehensive user manual and source code have been published beforehand and are referenced in the manuscript. We also provide examples for the application of the CCPP in two different host models in Section 4. We therefore believe that the manuscript fits the requirements for a "Model description paper" in GMD. In addition, the answers to the comments of reviewer 1 above, as well as the answers to this reviewer's comments below and in the revised manuscript do provide more background to the reader.

2. **Scientific aspects.**

   The reviewer combines many interesting questions into one paragraph regarding the flexibility of the CCPP Framework, which we try to group and answer in the following.

   - Sequential/parallel attitude. The current CCPP Framework has no explicit support for time-step (sequential) versus process-split (parallel) execution of physical parameterizations. However, the physics in use by both host models presented in the manuscript in Section 4 (Unified Forecast System UFS and CCPP SCM) contain both

time-split and process-split physics. The way this is realized in the current CCPP Framework is that time-split schemes ingest a model state in the form of several state variables, which they update in place. Process-split schemes ingest an input state and tendencies, and return updated tendencies. Work is underway to provide full support for time-split and process-split schemes, which requires extending the syntax of Suite Definition Files, updating the metadata standard, and augmenting the code generator to intercept state updates or translate tendencies into state updates as required.

- Order of schemes. Changing the order of schemes in CCPP is possible, as long as the physical parameterizations and in particular the interstitial schemes are written in a way that does not make assumptions of what is being called when. One real-world example from the UFS is that some of its suites call longwave radiation before shortwave radiation, whereas others call shortwave radiation first. Another, related example is the ability to remove the deep convection scheme from a Suite Definition File when running the model at a sufficiently high resolution.

- Inter-timestep history/implicit use/… The following set of questions all refer to the capability of calling CCPP schemes multiple times using subcycling and to call groups of schemes from different places in the model, as described in the manuscript. These two features allow the host model to perform the operations described by the reviewer. A real-world example is the implementation of the "fast physics" in the UFS, i.e. the call to the saturation adjustment for the NOAA-GFDL microphysics directly from the FV3 dynamical core within the acoustic loop. Likewise, implicit methods in the host model can be realized by calling the same CCPP group (which can be a group of one scheme) with the appropriate input state and timestep arguments. Within a particular scheme, it is the decision of the scheme developer which time integration to use for the scheme. Regarding inter-timestep history, it is possible to average physical processes over certain time ranges, but it would not follow the Lagrangian path to maintain the best spatial and temporal consistency.

- Super-parameterizations. In the current CCPP framework there is no explicit, formalized support for super-parameterizations. A host model that uses super-parameterization for a particular process/scheme needs to place this scheme into a separate group in the Suite Definition File and call this group separately after making the necessary modifications on the host model side (i.e. defining the higher resolution grid, updating the high-res state from the input state). After returning from the scheme, the host model must aggregate the updated state/apply the tendencies to the lower-resolution grid. Another possibility is that a CCPP scheme itself takes care of these steps, given a center coordinate and horizontal size of the grid column. The latter approach has its limitations, since the scheme has no information on the layout of the horizontal mesh and possibly no higher-resolution information (e.g. topography).

We added the following paragraph to Section 5 (Discussion and Outlook):
*"Modern NWP systems have vastly different technical requirements and use different*

*ways to call physical parameterizations, such as time-split versus process-split steps, where in the former a scheme updates a state before the next scheme is called, and in the latter multiple schemes operate on the same state before the combined tendencies are applied to update the state. The CCPP Framework currently has no explicit support for time-step (sequential) versus process-split (parallel) execution. However, both host models presented in Section 4 make use of time-split and process-split physics. This is realized such that time-split schemes ingest a model state (in the form of several state variables), which they update in place. Process-split schemes ingest an input state and tendencies, and return updated tendencies. Work is underway to provide full support for time-split and process-split schemes.*

*The ability of the CCPP Framework to call schemes multiple times using subcycling and to call groups of schemes from different places in the model allows host models to use implicit solvers or other higher-accuracy methods, with one example being to call the saturation adjustment for the GFDL microphysics scheme directly from the FV3 dynamical core within the fast, acoustic loop. The grouping of schemes also allows host models to implement super-parameterizations (Randall et al., 2013), i.e. the ability to call selected schemes with a higher-resolution grid than others, and piggybacking methods, i.e. "to run a single simulation applying two microphysical schemes, the first scheme driving the simulation and the second piggybacking this simulated flow" (Sarkadi et al., 2022)."*

3. **Technical questions, data storage/transfer between host model and CCPP**

It is a misunderstanding that CCPP always copies data before calling physics. The mechanism shown in Listing 1 is applied only if absolutely necessary, as in this example when combining blocked, non-contiguous data into contiguous arrays. We added the following text to the caption of Listing 1, which answers the latter half of Comment 2.

"*Note that data copy operations are only applied if absolutely necessary, as it is the case for converting blocked data into contiguous arrays, otherwise the host model variables are passed to the CCPP schemes directly.*"

Regarding the first half of Comment 2, it is correct that the current preferred storage model for CCPP schemes is targeted towards applications on CPUs rather than GPUs. However, the CCPP developers are fully aware of the move to GPUs in the NWP world and have therefore designed a mechanism to automatically transform arrays if the order of the dimensions doesn't match. This method has not been implemented yet, but will become available in a future version of the CCPP Framework. Note that Section 3.5.4 in the original manuscript already mentioned that the current implementation does not support automatic array transformations.

---

## Referee Report (RR1)

General Comments:

I find that most of my comments have been satisfactorily addressed in the revision. Although I have different opinions about DLL and used a different technical solution to implement our integration framework, I believe the revised manuscript can be published now after some confirmation. Many solutions can introduce advantages and disadvantages at the same time.

Specific comments:

1.  About Figure 2 and Listing 1.

    The code in Listing 1 just merges the data in different blocks into an array without following the parallel decomposition.

---

## Author Response (AR2)

**egusphere-2022-855: final response to minor revision 2023/03/24**

**Answer to Editor/Reviewer 2**

The authors thank reviewer 2 for the second review of our manuscript. The minor revision contained only one comment on Figure 2 and Listing 1:

1. *About Figure 2 and Listing 1.*
   *The code in Listing one just merges the data in different blocks into an array without following the parallel decomposition.*

Reviewer 2 is correct that the code in Listing 1 just merges the data in different blocks into an array without following the parallel decomposition. This does not impose any restrictions on physics operating on this data, since the physics in CCPP are per requirement column-based, i.e. independent of the horizontal decomposition. This is already explained in Section 3.1 of the manuscript. However, to provide further clarification, we added the following sentence to the end of Section 3.2 (bottom of page 7 in the tracked-changes manuscript):

*The combination of blocked data structures into contiguous arrays does not take into account the parallel decomposition of the data, since physical parameterizations in CCPP are by definition one-dimensional, i.e. independent of the horizontal decomposition.*